# Molecular basis of dimer formation during the biosynthesis of benzofluorene-containing atypical angucyclines

Chunshuai Huang[1,2], Chunfang Yang[1], Wenjun Zhang[1], Liping Zhang[1], Bidhan Chandra De[1,2], Yiguang Zhu[1], Xiaodong Jiang[1,2], Chunyan Fang[1,2], Qingbo Zhang[1], Cheng-Shan Yuan[1], Hung-wen Liu[3] & Changsheng Zhang [1]

Lomaiviticin A and difluostatin A are benzofluorene-containing aromatic polyketides in the atypical angucycline family. Although these dimeric compounds are potent antitumor agents, how nature constructs their complex structures remains poorly understood. Herein, we report the discovery of a number of fluostatin type dimeric aromatic polyketides with varied C−C and C−N coupling patterns. We also demonstrate that these dimers are not true secondary metabolites, but are instead derived from non-enzymatic deacylation of biosynthetic acyl fluostatins. The non-enzymatic deacylation proceeds via a transient quinone methide like intermediate which facilitates the subsequent C–C/C−N coupled dimerization. Characterization of this unusual property of acyl fluostatins explains how dimerization takes place, and suggests a strategy for the assembly of C–C and C−N coupled aromatic polyketide dimers. Additionally, a deacylase FlsH was identified which may help to prevent accumulation of toxic quinone methides by catalyzing hydrolysis of the acyl group.

[1] Key Laboratory of Tropical Marine Bio-resources and EcologyGuangdong Key Laboratory of Marine Materia Medica, RNAM Center for Marine Microbiology, South China Sea Institute of Oceanology, Chinese Academy of Sciences, 164 West Xingang Road, Guangzhou 510301, China. [2] University of Chinese Academy of Sciences, 19 Yuquan Road, Beijing 100049, China. [3] Division of Chemical Biology and Medicinal Chemistry, College of Pharmacy and Department of Chemistry, University of Texas at Austin, Austin, TX 78712, USA. These authors contributed equally: Chunshuai Huang, Chunfang Yang. Correspondence and requests for materials should be addressed to H.-w.L. (email: h.w.liu@mail.utexas.edu) or to C.Z. (email: czhang2006@gmail.com)

Benzofluorene-containing atypical angucyclines are widely distributed in nature[1] with lomaiviticin A (1)[2], kinamycin A (2)[3], nenestatin A (3)[4], and fluostatins (FSTs, such as 4–6) as representative examples (Fig. 1)[5–11]. Members of this family of natural products display a diverse range of biological activities. A case in point is lomaiviticin A (1) which exhibits cytotoxicity at nanomolar to picomolar concentrations by inducing double-strand breaks in DNA and is currently under preclinical evaluation for antitumor treatment[12–14]. Since the $C_2$-symmetric homodimer lomaiviticin A (1) and the C1−C5′ asymmetric heterodimer difluostatin A (6) are more potent than their respective monomers[8,12,14], dimerization appears to be important to amplify the therapeutic potential of this class of compounds. There has thus been significant interest in the biosynthesis of these compounds given their structural complexity and pharmaceutical potency. Recent progress in this regard includes identification of the biosynthetic gene clusters for the kinamycins (alp)[15,16], the lomaiviticins (lom)[17,18], nenestatin A (nes)[4], and the fluostatins (fls)[8, 11]. More recent biochemical experiments have also demonstrated that the oxidative ring contraction during kinamycin biosynthesis is catalyzed by the flavin-dependent monooxygenase AlpJ[19–21] and that hydrolysis of the epoxide moiety is catalyzed by the α/β hydrolase Alp1U[3]. The enzyme Lom6 (homologous to Alp1U) from the lomaiviticin pathway has the same activity as well[3]. However, how the carbon–carbon (C–C) bond between the two monomers is formed in 1 (C2/C2′) and 6 (C1/C5′) is still unknown[8,17,18].

Enzyme-catalyzed C–C bond formation is a fundamental process characterizing biosynthetic pathways. Claisen- and aldol-type condensations are among the most common mechanisms of C–C bond formation in biological systems[22]; however, enzymes catalyzing C–C bond-forming reactions via alternative mechanisms are also abundant, and study of their functions has been an active area in natural product research[23,24]. For example, cytochrome P450 enzymes are known to catalyze many intramolecular and intermolecular C–C bond-forming reactions and thus are responsible for the generation of many dimeric structures in natural product biosynthesis[24–26]. However, the lom and fls gene clusters lack genes encoding P450 enzymes. This leads to an early hypothesis that the production of the C–C-coupled dimers lomaiviticin A (1) and difluostatin A (6) might be catalyzed respectively by the regulatory proteins Lom19 and FlsQ1 of the NmrA family[8,18]. This is because Lom19 and FlsQ1 are homologs of the enzyme ActVA-ORF4, which has been demonstrated

in vivo to be essential for C−C dimerization during the biosynthesis of actinorhodins[27,28]. Yet, it is also possible that the formation of dimers 1 and 6 is spontaneous (i.e., not enzyme-catalyzed), since a recent study of indoloterpenoid biosynthesis implies that its dimer formation may be non-enzyme catalyzed[29].

Here we show that the heterologous expression of the fls-gene cluster in Streptomyces albus J1074 results in the isolation of several FST heterodimers/trimers with diverse C–C and C–N-coupling patterns. To investigate the mechanisms of their formation, two α/β hydrolases, FlsH and Lom6, which are thought to be responsible for converting intermediates in the fluostatin biosynthetic pathways to precursors for the dimerization reaction, are found instead to be acyl hydrolases of acyl FSTs. Importantly, acyl FSTs are also found in this study to undergo spontaneous deacylation leading to the formation of various C–C and C–N coupled homodimers/heterodimers. These results provide strong evidence that no enzyme is necessary for the dimerization of FSTs and thus solve a mystery that has puzzled natural product chemists for a long time. The results and the mechanistic implications of these experiments are reported herein.

## Results

**Heterologous production of diverse FST analogues.** Previous experiments have shown that heterologous expression of the fls-gene cluster from Micromonospora rosaria SCSIO N160 in Streptomyces coelicolor YF11[30] led to the production of new FSTs under sea salt-dependent culturing conditions[8]. To further exploit this observation, the fls-gene cluster was introduced into three other heterologous hosts: S. albus J1074, Streptomyces lividans TK64, and Streptomyces pactum SCSIO 02999 XM47i[31]. It was found that each host exhibits different metabolite profiles, with S. albus J1074 as the most prolific producer of FST metabolites (Supplementary Tables 1 and 2; Supplementary Fig. 1). A total of twenty-one compounds were isolated from a 40 L culture of the recombinant S. albus strain including the fourteen known compounds FST C (4), F (5), D (7), J (8), G/H (9), L (10), prefluostatin (11), FST K, prekinamycin, pyrazolofluostatins A–C, rabelomycin and dehydrorabelomycin (Supplementary Fig. 2)[8–10].

Also identified from the cultures of S. albus J1074 were three FST analogues, isoprefluostatin (12), FST R (13), and FST S (14), along with three dimeric derivatives, difluostatins B–D (15–17), and a trimeric compound, trifluostatin A (18) (Fig. 2). Preliminary identification of these compounds rested on high-

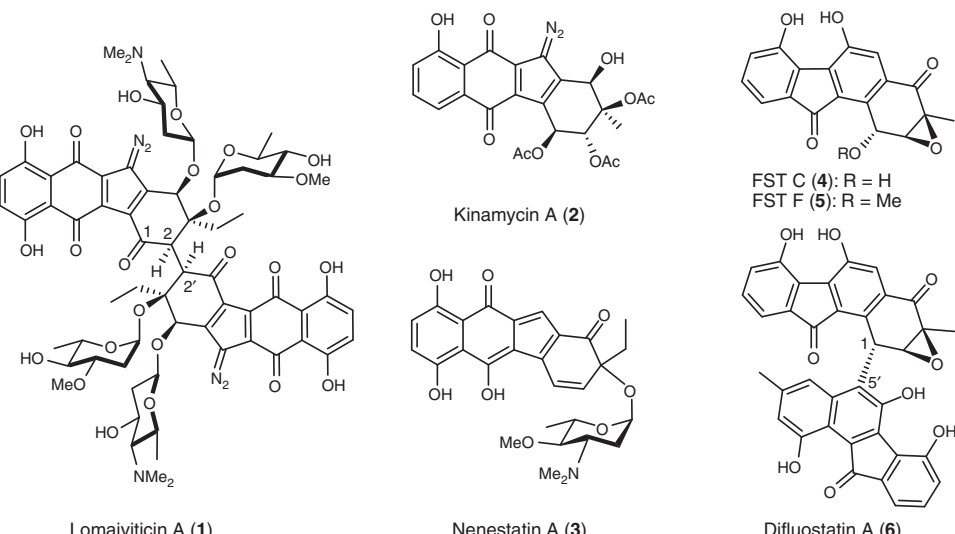

**Fig. 1** Representatives of atypical angucyclines containing a benzofluorene moiety

Lomaiviticin A (1)

Kinamycin A (2)

Nenestatin A (3)

FST C (4): R = H
FST F (5): R = Me

Difluostatin A (6)

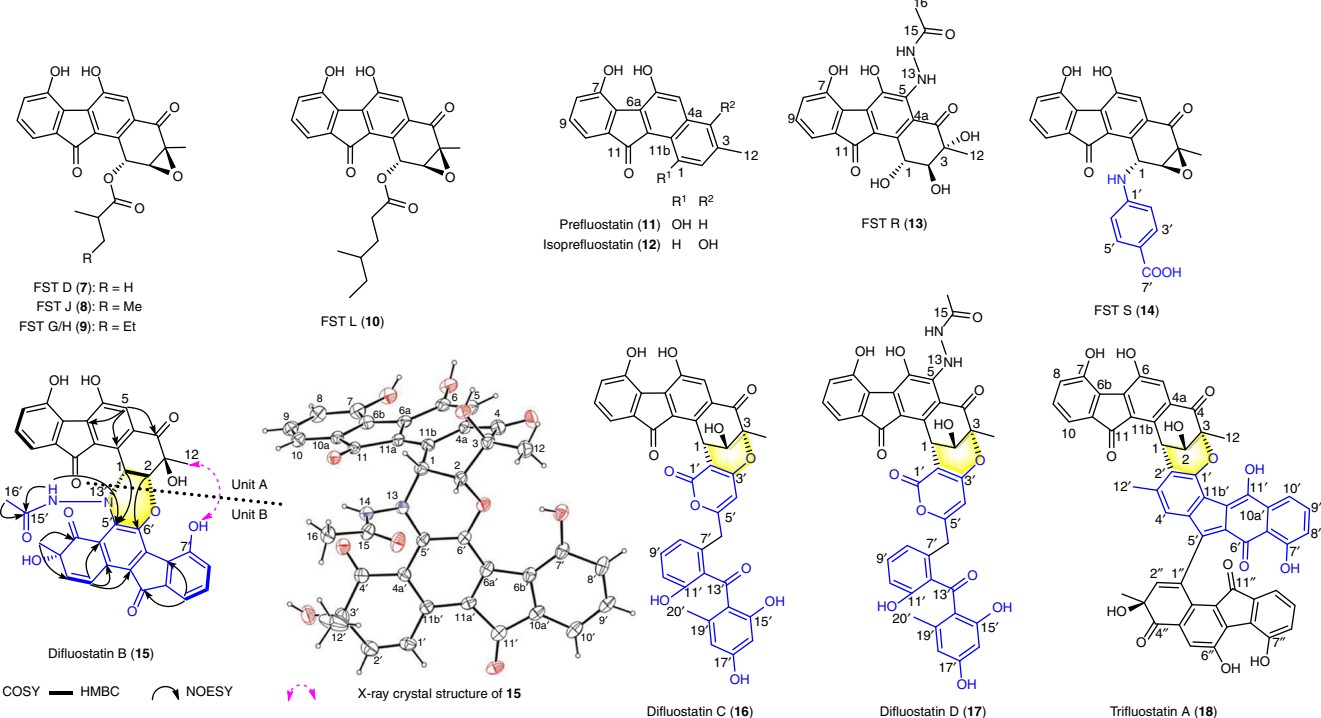

**Fig. 2** A partial list of FST-related metabolites isolated from the heterologous host *S. albus* J1074 harboring the *fls* gene cluster. Selected COSY, HMBC, and NOSEY correlations and the X-ray crystal structure of difluostatin B (**15**) are also shown. In dimeric compounds, the acceptors and donors are shown in black and blue, respectively. A full list of structures of compounds isolated in this study is provided in Supplementary Fig. 2

resolution electrospray ionization mass spectroscopy (HRESIMS) and nuclear magnetic resonance (NMR) (Supplementary Tables 3–5, Supplementary Figs. 3–50). Assignment of one of these compounds as difluostatin B (**15**) (*m/z* 685.1460 [M−H]⁻, calcd for $C_{38}H_{26}N_2O_{11}$: 685.1464), which is a heterodimer linked by a morpholine-like six-membered ring (Supplementary Table 4, Supplementary Figs. 23–29), was further supported by the observed coupling of the two expected units through C1–N13′–C5′ and C2–O–C6′ HMBC correlation patterns (Fig. 2) as well as NOESY correlation between 12-CH₃/7′-OH (Fig. 2). This assignment was subsequently confirmed by X-ray crystallographic analysis (Fig. 2; CCDC 1584739, Supplementary Table 6). Considering the biosynthesis context, difluostatin B (**15**) likely has the configuration of 1*R*, 2*R*, 3*S*, and 3′*S*. Unlike **15**, a pyran ring instead of a morpholine core is found in difluostatins C (**16**) and D (**17**), which are possibly formed via cross-coupling of a FST monomer (the acceptor) to a SEK43 (the donor) moiety[32]. A similar 6-membered pyran-ring core also exists in the structure of trimeric trifluostatin A (**18**) (Fig. 2). Interestingly, a C–N linkage is found in FST S (**14**) where a FST monomer (the acceptor) is joined with a *p*-aminobenzoic acid moiety (the donor). It was also noted that the coupling always occurs at the C1, C2, or C3 position of one of the monomeric units (the acceptor). The observation that heterologous expression of the *fls* gene cluster in a different host produced additional dimer/trimer products is particularly interesting. If their formation is catalyzed by enzyme(s) encoded in the *fls* gene cluster, the responsible enzyme(s) must have a fairly promiscuous substrate specificity for donors. The enzyme FlsQ1 was previously proposed to catalyze the coupling of FST C (**4**) and prefluostatin (**11**) to form difluostatin A (**6**)[8]. However, feeding of **4** and **11** to *E. coli* BL21 (DE3) expressing *flsQ1* did not show the production of **6** (Supplementary Figs. 51 and 52). In addition, the production of difluostatin A (**6**) was observed in the Δ*flsQ1* mutant where the *flsQ1* gene was inactivated by insertional mutagenesis

(Supplementary Figs. 53–55). Thus, FlsQ1 is unlikely the enzyme responsible for difluostatin dimer formation.

**Characterization of FlsH as a deacylase.** Since the FST acceptor monomer typically has an ester or an alcohol substituent at C1 and an epoxy or a diol moiety at C2 and C3 (Fig. 2), it is thus conceivable that cross-coupling with a donor molecule involves C–O bond cleavage and C–C/C–N bond formation at these loci through nucleophilic substitution reaction(s). To address the above hypothesis, we opted to determine whether hydrolysis of the epoxide at C2/C3 and/or the ester linkage at C1 is a prerequisite for the dimerization reaction. The opening of the epoxide ring in the biosynthesis of kinamycins and lomaiviticin A has been established to be catalyzed by Alp1U and Lom6, respectively, both of which are α/β hydrolases (Fig. 3a)[3]. An α/β hydrolase gene (*flsH*) also exists in the *fls* gene cluster. The encoded enzyme (FlsH) exhibits 37% and 60% sequence identity to Alp1U and Lom6, respectively (Supplementary Fig. 56). To examine if FlsH has similar epoxide hydrolyzing activity as Alp1U and Lom6, the *flsH*, *alp1U*, and *lom6* genes were individually overexpressed in *E. coli* and purified as soluble His₆-tagged proteins (Supplementary Fig. 57).

Upon incubation with Alp1U, FST C (**4**) was converted to two products with the same molecular mass (*m/z* 342) (Supplementary Fig. 58), which is 18 Da greater than that of **4**. These results are consistent with the generation of a pair of isomers **4a/4b** through addition of water at either C-2 (route a) or C-3 (route b) of **4** (Fig. 3a, b). In contrast, no turnover was observed when FST C (**4**) was incubated with Lom6 or FlsH (Fig. 3b). Furthermore, Alp1U, Lom6, and FlsH all failed to process FST F (**5**) which carries an *O*-methoxyl group at C1 (Fig. 3b). Surprisingly, while no reaction was detected when Alp1U was incubated with FSTs containing an *O*-acyl group at C1 (FSTs **7**−**10**), reactions of Lom6 and FlsH with these acyl FSTs led to the production of FST C (**4**) as the sole product, which was confirmed by co-elution with

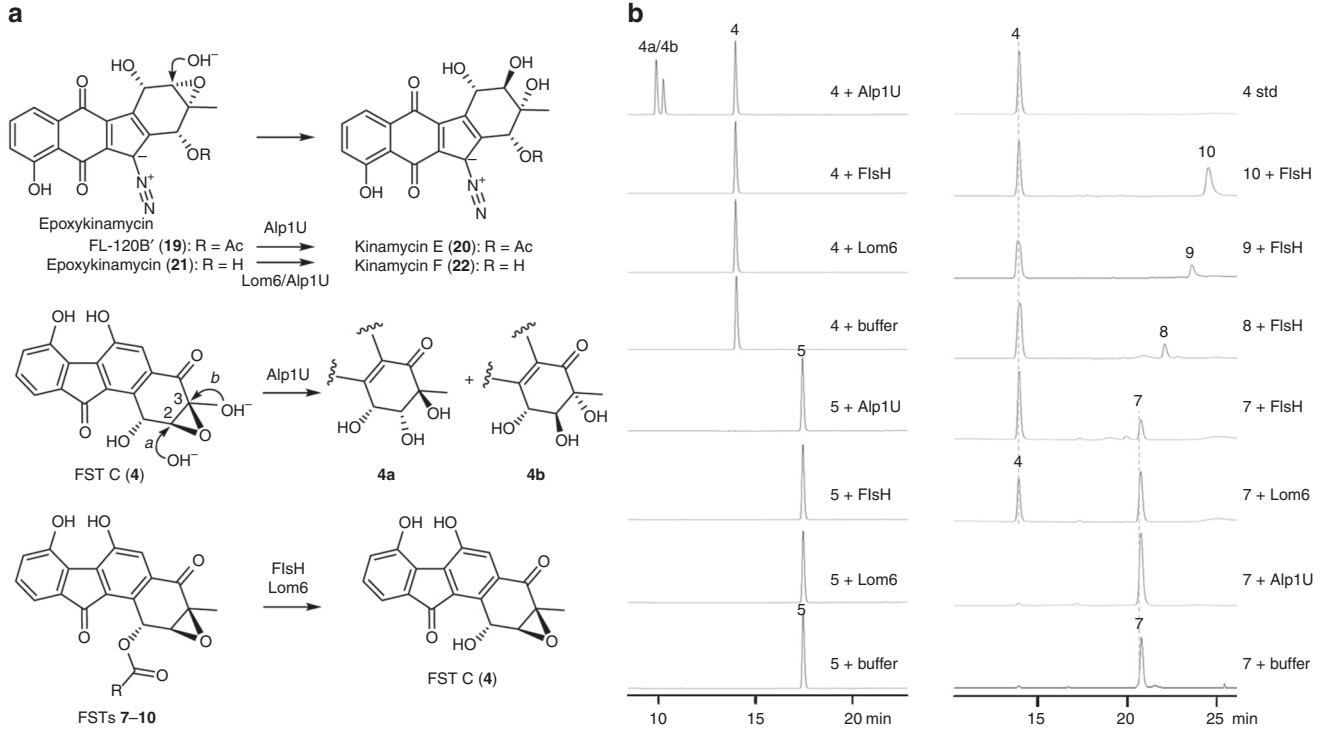

**Fig. 3** In vitro characterization of Alp1U, Lom6, and FlsH. **a** Scheme for Alp1U, FlsH, and Lom6 catalyzed reactions. **b** HPLC analysis of in vitro reactions of Alp1U, FlsH, and Lom6. Each enzyme assay was performed in 100 μL of 50 mM phosphate buffer (pH 7.0) containing 100 μM substrate (**4**, **5**, or **7–10**) and 10 μM enzyme (Alp1U, Lom6 or FlsH) for 30 min at 30 °C

the standard (Fig. 3b, Supplementary Fig. 59). Thus, unlike Alp1U which only functions as an epoxide hydrolase[3], Lom6 is a dual function enzyme capable of catalyzing epoxide hydrolysis in kinamycin biosynthesis[3] and the deacylation of acyl FSTs (Fig. 3a). Most importantly, FlsH was demonstrated to be a deacylase catalyzing the hydrolysis of the O-acyl group in FSTs **7–10**.

Homology modeling of Alp1U with the crystal structure of a well-known epoxide hydrolase from *Agrobacterium radiobacter* AD1 (PDB ID: 1EHY [https://www.rcsb.org/structure/1EHY][33]) shows a similar active site consisting of the catalytic residues (Asp137, His300, and Asp278) along with Tyr247 and Trp72 (Supplementary Fig. 60), which are all critical for epoxide hydrolyzing activity[33]. However, equivalent residues are not found in FlsH, which may explain why FlsH could not hydrolyze an epoxide moiety. The structure model of FlsH is built according to a template structure of an α/β hydrolase from *Sphaerobacter thermophilus* DSM 20745 (PDB ID: 3R0V [https://www.rcsb.org/structure/3R0V]), which shares 45% sequence identity to FlsH (Supplementary Fig. 61). Docking of FST J (**8**) into the structural model of FlsH reveals the presence of the catalytic triad Ser92, His241, and Glu115 (Fig. 4a) commonly seen in classical serine proteases/esterases (Ser-His-Asp/Glu triad)[34]. Indeed, all three FlsH mutants S92A, E115A, and H241F lost their deacylation capability toward FST J (**8**) (Fig. 4b). Hence, the deacylation reaction catalyzed by FlsH is believed to operate based on a mechanism analogous to that of typical serine-esterases (Supplementary Fig. 62).

**Spontaneous deacylation and accompanied formation of dimers.** To further characterize FlsH, a time course analysis of its catalyzed deacylation of FST D (**7**) was carried out. The results showed that the deacylation of **7** to FST C (**4**) was nearly completed within 1 h (Supplementary Fig. 63). A parallel assay without FlsH was also performed as a control (Supplementary

Fig. 63); interestingly, in this control FST D (**7**) was also consumed albeit at a slower rate (nearly complete after 12 h, Fig. 5a). In addition to FST C (**4**), two additional products identified as **23** and **24** that were not observed in FlsH-catalyzed reaction were also detected (Fig. 5a). Similar results were noted as well upon incubation of FST J (**8**) in H₂O without FlsH. In this case, FST C (**4**), **24**, and **25** were obtained as products (Fig. 5a). Compounds **23–25** are C1–C10′-coupled FST-dimers (Fig. 5b, Supplementary Figs. 64–78, Supplementary Table 7), designated difluostatins E–G. Incubation of FST D (**7**) and FST J (**8**) in MeOH afforded FST F (**5**) and a minor product characterized as difluostatin H (**26**) (Fig. 5a, b; Supplementary Fig. 79–85, Supplementary Table 7), whereas non-acylated FSTs, such as FSTs C (**4**) and F (**5**), were stable under identical conditions. The spontaneous deacylation of FST J (**8**) in H₂O was found to proceed with a first order rate constant ($k_{non}$) of approximate 0.003 min⁻¹ (Supplementary Fig. 86). In contrast, the deacylation of FST J (**8**) mediated by FlsH was determined to have a $k_{cat}$ of 0.70 min⁻¹, a $K_m$ of 21.06 μM, and a $k_{cat}/K_m$ of 0.033 μM⁻¹ min⁻¹ (Supplementary Fig. 86). Thus, FlsH-catalyzed deacylation displays an almost 230-fold rate enhancement compared to the spontaneous reaction. It is worth mentioning that the spontaneous elimination of an O-acetyl group to form a double bond was previously observed in the biosynthesis of fungal natural product anditomin[35]. However, its mechanism had never been studied.

**Mechanistic insights into non-enzymatic reactions.** While the mechanism of FlsH-catalyzed deacylation is presumably similar to that of the serine-esterases, the mechanism of spontaneous deacylation is expected to differ from that of classical ester hydrolysis reactions, because the reaction is accompanied by the production of dimeric FSTs such as **23−26**. It is conceivable that the spontaneous deacylation may generate an electrophilic intermediate (from the acceptor) which is susceptible to

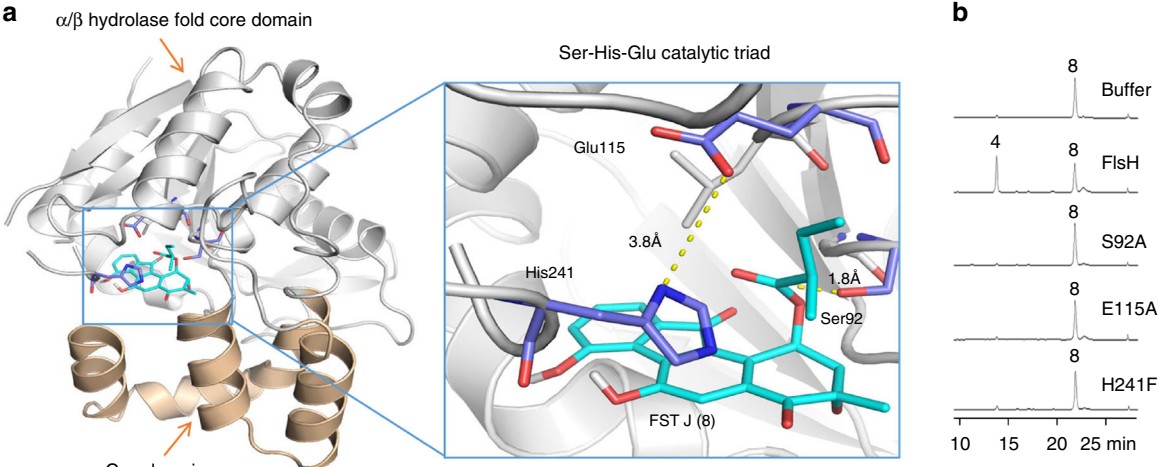

**Fig. 4** Identification of the catalytic triad in FlsH. **a** A structure model of FlsH highlighting the catalytic triad (Ser-His-Glu) and the docking with FST J (**8**). **b** HPLC analysis of enzyme assays with FlsH and three mutated variants. Each enzyme assay was performed in 100 μL of 50 mM phosphate buffer (pH 7.0) containing 100 μM **8** and 10 μM enzyme (FlsH or mutated variants S92A, E115A or H241F) for 30 min at 30 °C

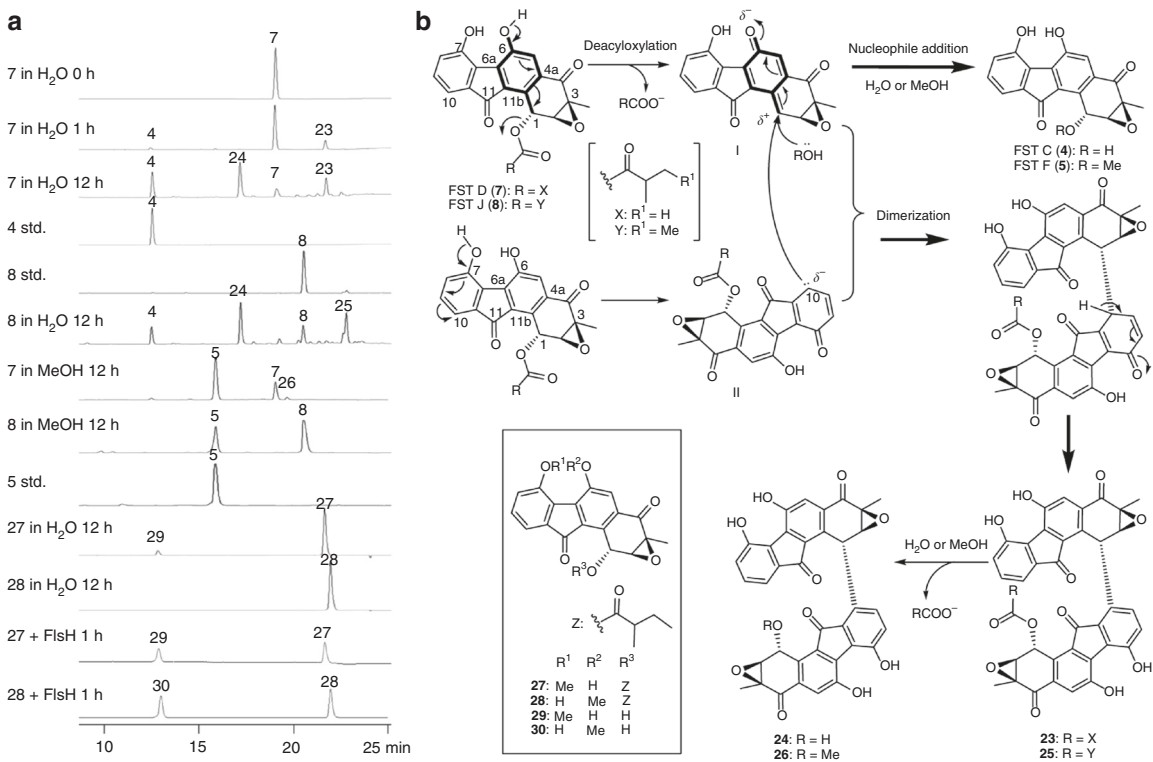

**Fig. 5** Analysis of non-enzymatic reactions of acyl FSTs and the proposed two-step mechanisms. **a** HPLC analysis of non-enzymatic reactions of acyl FSTs. The general conditions for incubation were set as: substrate (**7**, **8**, **27** or **28**) 100 μM, enzyme 10 μM, 30 °C. **b** Proposed mechanisms for the spontaneous deacylation and accompanied dimerization. Chemical structures of compounds **27**−**30** are shown in a block as an inset

nucleophilic addition by another monomer (the donor) in the subsequent step to form the dimeric product. Careful inspection of the structures of acyl FSTs reveals the presence of a built-in *para*-hydroxyl benzyl framework (highlighted in bold, Fig. 5b). Since *p*-hydroxyl benzyl acetate is known to undergo spontaneous acetyl elimination to yield a *p*-quinone methide (*p*-QM) product which is a Michael acceptor[36], formation of a *p*-QM-like intermediate from the built-in *p*-hydroxyl benzyl moiety can thus account for the observed deacylation as well as dimer formation in the spontaneous reaction.

To test this hypothesis, FST J (**8**) was treated with trimethylsilyldiazomethane (TMSCHN₂) to give two monomethylated products **27** and **28** (Fig. 5b, Supplementary Figs. 87–100, Supplementary Table 8). Compound **27** bearing a C7 OMe group could still undergo spontaneous deacylation to yield a single product (Fig. 5a), the retention time and mass of which were consistent with the synthetic standard of **29** (Supplementary Figs. 101–107, Supplementary Table 9). No dimer formation was discernible in this case (Fig. 5a). On the contrary, compound **28** which carries a C6 OMe group was not susceptible to either

deacylation or dimer formation (Fig. 5a). These results indicated that the free hydroxyl group at C6 in acyl FSTs is essential for spontaneous deacylation, and the free hydroxyl group at C7 is necessary for dimerization. Both **27** and **28** were still substrates for FlsH being hydrolyzed to yield **29** and **30** (Fig. 5a; Supplementary Figs. 108–114, Supplementary Table 9), respectively.

Taken together, the spontaneous deacylation of acyl FSTs likely proceeds via a two-step reaction mechanism (Fig. 5b). The first step is a 1,6-elimination process leading to a transient $p$-QM intermediate (**I**) by deacyloxylation (Fig. 5b). This is followed by the nucleophile attack on **I** with $H_2O$ or MeOH to generate the deacylated product FST C (**4**) or FST F (**5**). Alternatively, the nucleophile in the second step could be another FST (such as **7** or **8**) whose C10 is nucleophilic due to conjugation with the C7 OH substituent (see **II** in Fig. 5b). The intermolecular nucleophilic attack from the electron rich C10 of **II** to the electron deficient C1 of **I** could generate a C1−C10′ dimer (such as **23** and **25**, Fig. 5b), which after a similar two-step process affords **24** (in water) or **26** (in MeOH) as the product.

The proposed mechanism (Fig. 5b) is further supported by the following facts: (i) acyl FSTs (such as **7**) are quite stable in aprotic organic solvents, e.g., dimethyl sulfoxide (DMSO), acetone, and chloroform (Supplementary Fig. 115); (ii) FST D (**7**) is also stable under acidic conditions (almost no change in buffers with pH lower than 4.0), but is readily converted to dimeric FSTs under basic conditions (Supplementary Fig. 116). This is likely due to the ease of deprotonation of both phenol groups at C6 and C7 under basic conditions to facilitate the formation of the $p$-QM intermediate **I**, which is the precursor for the subsequent dimerization reaction; (iii) compound **28** is inert to deacylation, which is consistent with the electron-donating tendency for a $p$-phenoxide ($O^-$) ($\sigma_p = -0.81$) versus a $p$-$OCH_3$ group ($\sigma_p = -0.27$) as indicated by their Hammett constants[37]; (iv) $^{18}O$ is incorporated into **4** and **24** when the reaction is conducted in $H_2^{18}O$, which rules out a mechanism of simple hydrolysis for the deacylation reaction (Supplementary Fig. 117).

**The utility of deacylation-triggered reactions.** Based on the above results, difluostatin A (**6**), which was previously proposed to be generated from an enzyme-catalyzed carbon–carbon bond formation reaction[8], may instead be produced through coupling between two FSTs after the spontaneous deacylation of the acceptor monomer (Fig. 6a). Indeed, formation of **6** was observed when prefluostatin (**11**) and FST D (**7**) were incubated for 12 h in water (Fig. 6a, Supplementary Fig. 118). Thus, difluostatin A (**6**) is not a real secondary metabolite but merely a pseudo natural product because its formation is a post-biosynthesis non-enzymatic event. Not only C–C bond but also C–N bond coupling is possible as exemplified by the formation of **14** via co-incubation of **7** with $p$-aminobenzoic acid (PABA, **31**) (Fig. 6a, Supplementary Fig. 119). Furthermore, a new product **33** containing a seven-membered 1,4-oxazepane-like core was observed when **7** was treated with 2-amino-5-methylphenol (**32**) (Fig. 6a; Supplementary Figs. 120−127, Supplementary Table 10). The reaction must be a result of an initial C−N coupling followed by a C1′-OH-mediated epoxide opening at C3 (Supplementary Fig. 120).

These findings suggested that the $p$-QM intermediates derived from acyl FSTs hold promise to couple with a variety of nucleophiles to make FST-conjugates[38]. A proof of concept experiment was carried out in which FST D (**7**) was incubated with the antibacterial agent trimethoprim (TMP, **34**). Two products **35** and **36** isolated from this reaction (Fig. 6a, Supplementary Fig. 128) were structurally characterized as hybrids of TMP and FST (Fig. 6a; Supplementary Figs. 129−142). Similar

results were obtained upon incubation of TMP (**34**) with compound **27**, but not with compound **28** (Supplementary Fig. 143). These findings again support the hypothesis that spontaneous formation of a $p$-QM precursor is a prerequisite for the production of FST hybrids. Unfortunately, none of the isolated new products showed significant antimicrobial activities against seven indicator strains, including *Staphylococcus aureus* ATCC 29213, *Escherichia coli* ATCC 25922, *Enterococcus faecalis* ATCC 29212, *Acinetobacter baumannii* ATCC 19606, *Bacillus subtilis* SCSIO BS01, *Micrococcus Luteus* SCSIO ML01, and methicillin resistant *S. aureus* ATCC 43300, with MIC (minimal inhibition concentration) values greater than $16\,\mu g\,mL^{-1}$ (Supplementary Table 11).

**Discussion**
A large number of dimeric natural products have been isolated but how they are biosynthesized in most cases are not known. While the assembly of some of them have been proposed to involve [4 + 2] or [2 + 2] cycloaddition[39–44], whether the reactions are actually enzyme-catalyzed remain to be verified. Among the limited examples where dimerizations have been shown to be enzyme-mediated, the reactions are commonly catalyzed by cytochrome P450 enzymes as in the cases of himastatin (C5−C5′)[45], ditryptophenaline (C3−C3′)[46], julichrome Q (C7−C7′)[47], hexahydroxyperylene-quinone (C4−C4′)[48], polybrominated biphenyls (C1−C1′)/dipyrroles (C2−C2′)/phenyl–pyrroles (C1−C2′)[49], desertorins (C6−C8′), orlandin (C8−C8′)[50], cladofulvin (C4−C2′)[51], and the dimeric flaviolins (C3−C3′ or C3−C8′)[52] (Supplementary Fig. 144). Other modes of biological dimerization are also known, such as CylK-catalyzed aromatic ring alkylation leading to head-to-tail dimerization of a halogen-activated alkyl resorcinol intermediate in the biosynthesis of cylindrocyclophanes[53]. The NmrA family of regulatory proteins, such as ActVA-ORF4, Lom19, and Strop2191, were also implicated as the C−C dimerization enzymes in the construction of actinorhodins[27,28] and lomaiviticins[17,18].

In this study, we have demonstrated that formation of the FST-type dimers is not enzyme-catalyzed. They are instead formed via an autocatalytic 1,6-elimination of the acyl group in FSTs (**7–10**) to yield a reactive $p$-QM-like intermediate which then undergoes coupling with a nucleophilic donor to produce diverse C−C and C−N-linked homo-/hetero-dimeric FSTs under mild conditions (Fig. 5b). The nucleophilic coupling to a $p$-QM is governed by the HOMO-LUMO (the highest energy occupied molecular orbital—the lowest energy unoccupied molecular orbital) interactions between the incoming nucleophile and the qunione methide. The commonly observed $\delta$-addition of nucleophile to the $p$-QM system is both kinetically and thermodynamically more favorable than the $\beta$-addition[54]. The QM moiety has been proposed to be a biosynthetic intermediate in the assembly of some natural products[55–57]. A recent example is the identification of elansolid A3, a $p$-QM-containing metabolite, which is the key intermediate in the biosynthesis of elansolides in *Chitinophaga santi*[58,59]. A variety of natural products have also been shown to produce $p$-QM-like products either spontaneously or after enzymatic transformation[54,60]. The reactive $p$-QM species could then react with various nucleophiles to yield adducts of diverse structures[61,62]. An early example is the elimination of the C7 $O$-glycoside of daunomycin to enable the preparation of a number of C7 thiol substituted daunomycin derivatives[61].

The autocatalytic formation of $p$-QMs from acyl FSTs can account for the production of many *pseudo* natural products identified in our work. Since FST D (**7**), after deacylation, could couple with 2-amino-5-methylphenol (**32**) to form **33**, the morpholine-like six-membered ring in difluostatin B (**15**) is

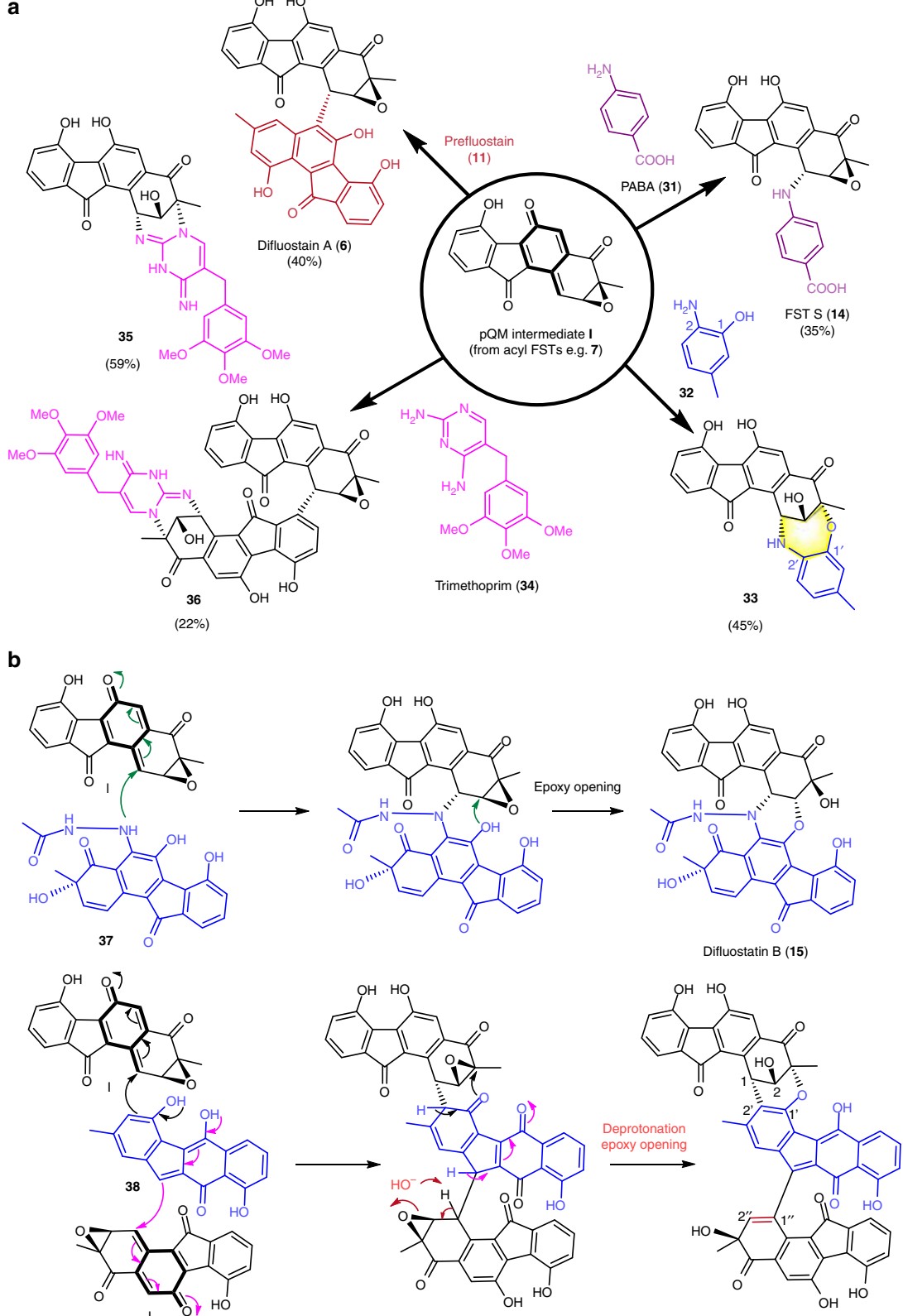

**Fig. 6** Synthesis of FST derivatives utilizing *p*-QM chemistry and the mechanistic implications. **a** Synthesis of diverse FST derivatives from FST D (**7**). Difluostain A (**6**) was synthesized by co-incubation of **7** and **11**; FST S (**14**) was synthesized by co-incubation of **7** and PABA (**31**); compound **33** was synthesized by co-incubation of **7** and 2-amino-5-methylphenol (**32**); **35** and **36** were synthesized by co-incubation of **7** and trimethoprim (**34**). For each reaction, an incubation in H₂O was performed overnight at room temperature. The conversion rate of each compound is indicated in parenthesis. **b** The proposed *p*-QM (**I**)-mediated spontaneous formation of difluostatin B (**15**) and trifluostatin A (**18**)

therefore expected to be generated in an analogous manner by a deacyloxylation-triggered coupling of a *p*-QM intermediate (**I**) with a not-yet-isolated FST congener **37** through C−N formation, followed by a C6′ OH-mediated epoxide opening at C2 (Fig. 6b). Likewise, trifluostatin A (**18**), an FST trimer, can be produced from coupling of two *p*-QM species (**I**) with benzuofluorene **38** (Fig. 6b), which is a product of AlpJ-catalyzed ring contraction reaction[19–21]. The olefinic bond between C1″ and C2″ in tri-fluostatin A may be formed via an intermediate, which undergoes a deprotonation-triggered epoxide opening under basic conditions[63]. Difluostatins C and D (**16** and **17**) could also be artifacts resulted from dimerization of the *p*-QM intermediate (**I**) with SEK43 (Supplementary Fig. 145). In view of the absence of type II PKS gene clusters in the genome of *S. albus* J1074[64], the SEK43 moiety in **16** and **17** must be derived from an aberrant cyclization process catalyzed by the type II PKSs encoded in the *fls* gene cluster (Supplementary Fig. 145).

While acyl FSTs are precursors of *p*-QMs, these acyl FSTs are also substrates for the α/β hydrolases FlsH and Lom6 capable of hydrolyzing the acyl group of acyl FSTs (Fig. 5b). Since *p*-QMs could react with DNA, proteins and other cellular targets, their formation may potentially be detrimental to the cells[60]. Therefore, the occurrence of a deacylase (such as FlsH) in the FST pathway may be necessary to control the physiological concentrations of acyl FSTs to minimize the possible formation of harmful *p*-QM-like molecules.

In conclusion, a number of FST-type aromatic polyketides with diverse C−C and C−N coupling patterns were discovered, and the dimeric structures of FSTs are found not to be true secondary metabolites but are derived from coupling of various nucleophilic donors to the *p*-QM intermediates generated via non-enzymatic deacylation of appropriate acyl FSTs. Furthermore, a deacylase FlsH was characterized which may be evolved in the FST pathway to prevent the accumulation of toxic *p*-QMs by enzymatic hydrolysis of the acyl group. Importantly, the *p*-QM intermediates were demonstrated to be useful for generating FST dimers, and making FST-conjugates with other bioactive compounds. Finally, our results highlight the importance of perceiving structure isolation/determination of natural products from a biosynthetic point of view because many of them could be artifacts if their assembly is non-enzyme catalyzed.

## Methods

**General**. General materials and methods are summarized in Supplementary Methods. Bacteria strains and plasmids used and constructed in this study are summarized in Supplementary Table 1. Primers used in this study are listed in Supplementary Table 2.

**Heterologous expression and compounds isolation**. The recombinant plasmid pCSG5033[8] containing the *fls*-gene cluster (Supplementary Table 1) was introduced into *S. albus* J1074, *S. lividans* TK64, *S. pactum* SCSIO 02999 XM47i, and *S. coelicolor* YF11 (Supplementary Table 1), respectively, for heterologous expression. Three exoconjugants were randomly selected to be incubated in 50 mL of production media (0.1% peptone fish, 1% starch soluble, 0.6% corn powder, 0.2% bacterial peptone, 0.5% glycerol, 0.2% CaCO$_3$, 3% sea salt, pH 7.0) and cultured at 28 °C for 4–7 days. The corresponding strains harboring the vector pSET152 (Supplementary Table 1) were treated in the same manner as controls. The fermentation of *S. albus* J1074/pCSG5033 was performed in the production media of a total volume of 40 L at 28 °C for 7–8 days. The crude extracts of fermentation were subjected to diverse chromatographic steps including silica gel (100–200 mesh) column chromatography, preparative thin-layer chromatography (TLC), Sephadex LH-20 column chromatography, C18 reversed phase medium pressure liquid chromatography (MPLC, 40 × 2.5 cm ID), MCI gel column chromatography, and reversed-phase semi-preparative high performance liquid chromatography (HPLC) to afford FSTs: isoprefluostatin (12, 8.0 mg), FST R (13, 0.5 mg), FST S (14, 4.0 mg), difluostatins B–D (15, 7.3; 16, 54.0; 17, 9.0 mg), and trifluostatin A (18, 2.8 mg). The reversed-phase semi-preparative HPLC was performed on a Hitachi-L2130 station (equipped with a Hitachi L-2455 diode array detector) using a Phenomenex C18 column (250 mm × 10 mm, 2.5 mL min$^{-1}$) under the following program: solvent system (solvent A, 10% acetonitrile in water supplementing with 0.08%

formic acid; solvent B, 90% acetonitrile in water); 5% B to 80% B (0–20 min), 80% B to 100% B (20–21 min),100% B (21–24 min), 100% B to 5% B (24–25 min), 5% B (25–30 min); flow rate at 2.5 ml min$^{-1}$, UV detection at 304 nm.

**X-ray crystallographic analysis**. An optically active red crystal of **15** was obtained in MeOH/H$_2$O. The crystal data were recorded on a Rigaku XtaLab PILATUS3 R 200 K diffractometer with Cu Kα radiation (λ = 1.54184 Å). Crystallographic data have been deposited in the Cambridge Crystallographic Data Center with the deposition number CCDC 1584739. A copy of the data can be obtained, free of charge, on application to the Director, CCDC,12 Union Road, Cambridge CB21EZ, U.K. (fax, + 44(0)-1233-336033; e-mail, deposit@ccdc.cam.ac.uk).

**Co-incubation of 4 and 11 in *E. coli* expressing *flsQ1***. The *flsQ1* gene was amplified by PCR from the genomic DNA of *M. rosaria* SCSIO N160, and was cloned into pET28a to afford pCSG5209 (Supplementary Tables 1 and 2). Expression of *flsQ1* was induced by the addition of 0.1 mM isopropyl-β-D-thio-galactopyranoside (IPTG) into *E. coli* BL21(DE3)/pCSG5209 growing at 16 °C to an A$_{600}$ of around 0.7. After further incubation for 3 h, both compounds FST C (**4**) and prefluostatin (**11**) were added to the culture for additional incubation of 20 h. The products were then extracted with butanone and analyzed by the HPLC. *E. coli* BL21(DE3) harboring pET28a was treated in the same manner as a control.

**Insertional mutagenesis of *flsQ1***. The *flsQ1* gene was inactivated by PCR-targeting method using the apramycin resistance cassette[8]. Details for the insertional inactivation of *flsQ1* are described in Supplementary Fig. 53. Conjugation between *E. coli* ET12567/pUZ8002/pCSG5017 (Supplementary Table 1) to *M. rosaria* SCSIO N160 was performed using the following method[8]. Specifically, the cell pellets of *E. coli* ET12567/pUZ8002/pCSG5017 were gently mixed with mycelia of *M. rosaria* SCSIO N160, and then the mixtures were then plated on the ISP4 solid medium. After incubation at 30 °C for 20–24 h, the plates were supplemented with the antibiotics apramycin (100 μg mL$^{-1}$) and trimethoprim (TMP, 100 μg mL$^{-1}$) for the selection of positive transconjugants.

**Protein expression and purification**. The *flsH* gene was PCR amplified from the genomic DNA of *M. rosaria* SCSIO N160, and the PCR product was cloned into pET28a to yield the plasmids pCSG5213 (Supplementary Tables 1 and 2). The DNA fragment of *lom6* from the lomaiviticin biosynthetic gene cluster in *Salinispora pacifica* strain DPJ-0019 (GenBank accession no. KF515737 [https://www.ncbi.nlm.nih.gov/nuccore/KF515737]) and the *alp1U* gene from the kinamycin biosynthetic gene cluster in *S. ambofaciens* ATCC 23877 (GenBank accession no. AM238663 [https://www.ncbi.nlm.nih.gov/nuccore/AM238663]) were synthesized and cloned into pET28a to afford plasmids pCSG5226 and pCSG5225 (Supplementary Table 1). Site-directed mutagenesis of FlsH (S92A, E115A, H241F) (Supplementary Table 1) was carried out according to manufacturer's instructions (TransGen) and the mutations in the constructs were confirmed by sequencing.

The plasmid pCSG5213 was introduced into *E. coli* BL21(DE3) for overexpression of *flsH*. When the cultures in LB media containing kanamycin (50 μg mL$^{-1}$) were grown to an OD$_{600}$ of 0.6 at 37 °C, the production of FlsH was induced by the addition of IPTG to a final concentration of 0.1 mM. The cultures were grown at 16 °C for an additional 20 h. The cells were then collected by centrifugation and were resuspended in the lysis buffer (20 mM Tris-Cl, 500 mM NaCl, and 5 mM imidazole, pH 8.0) for sonication. Purification of His-tagged recombinant FlsH was conducted using Ni-NTA affinity chromatography according to the manufacturer's manual (Novagen, USA). After desalting with PD-10 column (GE Healthcare, USA), the purified FlsH was stored in the storage buffer (10% glycerol, 1 mM DTT, 50 mM Tris-Cl, 100 mM NaCl, pH 8.0) at −80 °C for further use. The recombinant proteins of Lom6, Alp1U, and FlsH mutants (S92A, E115A, H241F) were prepared using the same method.

**In vitro enzyme reactions**. A typical in vitro enzyme reaction (Alp1U, FlsH, and Lom6) was conducted in 100 μL phosphate buffer (50 mM, pH 7.0) comprising 100 μM substrate (**4**, **5**, **7–10**, **27**, or **28**), 10 μM enzyme (Alp1U, FlsH or Lom6). The reactions were incubated at 30 °C for 30 min and then were quenched by mixing with 50 μL of ice-cold MeOH. In a time course assay of the FlsH-catalyzed reaction of **7**, samples were taken at 0, 10, 30, 60, 120, and 180 min (Supplementary Fig. 63). HPLC analysis of the enzyme reactions was carried out on the Agilent 1260 Infinity series instrument (Agilent Technologies Inc., USA) using a reversed phase column Luna C18 (5 μm, 150 × 4.6 mm, Phenomenex) with UV detection at 446 nm under the following program: solvent system (solvent A, 10% acetonitrile in water supplemented with 0.1% formic acid; solvent B, 90% acetonitrile in water); 5% B to 80 % B (0–20 min), 100% B (21–24 min), 100% B to 5% B (24–25 min), 5% B (25–30 min); flow rate at 1 ml min$^{-1}$.

**Protein homology modeling and in silico docking**. The structure models of FlsH and Alp1U are built by I-Tasser online server[65]. FST J (**8**) is docked into the deduced FlsH active site using AutoDock Vina[66].

**Non-enzymatic reactions of acyl FSTs**. The non-enzymatic reactions of acyl FSTs were performed by incubation of **7** or **8** (100 μM) in different solvents (such as $H_2O$, MeOH, DMSO, chloroform, and acetone) overnight at 30 °C. In a time course assay of the stability of **7** in $H_2O$, samples were taken at 0, 10, 30, 60, 120, 180, 240 min, 8 h, and 12 h (Supplementary Fig. 63). In a time course assay of the pH-dependent stability of **7**, reactions were conducted by incubation of **7** (10 μM) at 30 °C in phosphate buffer saline (PBS) with pH values ranging from 3 to 10, and samples were taken at 1, 6, 12, and 15 h for HPLC analysis (Supplementary Fig. 116).

**Determination of kinetic data of deacylation reactions**. For determining the kinetic parameters of FlsH-catalyzed reaction, FST J (**8**) was used as the substrate with concentrations varied from 5, 10, 15, 30, 125, to 200 μM. Enzyme assays were performed in phosphate buffer (50 mM, pH 7.0) containing 0.5 μM FlsH, at 30 °C for 5 min in triplicates. Kinetic parameters ($K_m$, $k_{cat}$, $V_{max}$) were determined by nonlinear regression analysis using GraphPad Prism 6 software. For the non-enzyme catalyzed deacylation of **8**, a standard curve of the concentrations (on the X axis) versus the peak areas (on the Y axis) was first plotted (Supplementary Fig. 86). The incubation of **8** at a final concentration of 0.1 mM or 0.2 mM in water was performed at room temperature. Samples were taken at 0, 90, 180, 360, and 540 min and were analyzed by HPLC. Concentration of FST J (**8**) at each sampling time point was determined by calibration against the standard curve. The curve of incubation time versus the concentrations of **8** was obtained by linear fitting the experimental data. Linear correlation coefficients and equations were shown for each curve, and then the first order rate constant ($k_{non}$) for the nonenzymatic degradation of FST J (**8**) was determined (Supplementary Fig. 86).

**Synthesis of methylated derivatives 27−30**. The chemical methylation of FST J (**8**) or FST C (**4**) was performed using the following method[29]. TMSCHN₂, 2.0 M solution in hexanes, 0.32 mmol, 160 μL was slowly added to a solution of **8** (0.04 mmol, 16.0 mg) in MeOH (1.0 mL). After stirring the mixture for 12 h at room temperature, AcOH (0.34 mmol, 20 μL) was added, and the mixture was stirred for 15 min. The mixture was evaporated to dryness under reduced pressure, and the crude extract was subjected to semi-preparative HPLC using a Hitachi-L2130 HPLC equipped with a Hitachi L-2455 diode array detector, eluting with isocratic solvent system (70% acetonitrile in water supplementing with 0.08% formic acid; flow rate at 2.5 mL min$^{-1}$) to yield **27** (6.5 mg, 39%) and **28** (7.0 mg, 41%). Based on a similar procedure, **29** (5.8 mg, 43%) and **30** (6.2 mg, 46%) were prepared from FST C (**4**, 0.04 mmol, 13.0 mg).

**Synthesis of 6, 14, 33, 35, and 36 from FST D (7)**. Prefluostatin (**11**, 0.01 mmol, 3.0 mg) and FST D (**7**, 0.02 mmol, 8.0 mg) were mixed in water. After an overnight incubation at room temperature, the mixture was extracted with equal volume of EtOAc and the organic extracts were concentrated under vacuum to yield a crude extract. The crude extract was analyzed by HPLC and LC-MS. The product was produced in 40% yield, and the identity of which was confirmed by co-elution with the standard **6** (Supplementary Fig. 118). Similarly, an overnight co-incubation of PABA (**31**, 0.04 mmol, 6.0 mg) and **7** (0.01 mmol, 4.0 mg) in water at room temperature generated **14** in around 35% yield judging by HPLC analysis. Its identity was verified by MS analysis and comparison with the standard **14** (Supplementary Fig. 119). Compound **33** was prepared from coincubation of 2-amino-5-methylphenol (**32**, 0.16 mmol, 20.0 mg) and **7** (0.04 mmol, 16.0 mg) in water at room temperature overnight (Supplementary Fig. 120). The mixture was extracted with equal volume of EtOAc and concentrated under vacuum to give a crude extract. The crude extract was purified via semi-preparative HPLC to yield **33** (7.8 mg, 45%). In a similar manner, compounds **35** (14.0 mg, 59%) and **36** (4.0 mg, 22%) were prepared from an overnight reaction of trimethoprim (**34**, 0.16 mmol, 47.0 mg) and **7** (0.04 mmol, 16.0 mg) in water at room temperature (Supplementary Fig. 128).

**Antimicrobial activity**. Antimicrobial activities were measured against seven indicator strains (*Staphylococcus aureus* ATCC 29213, *Escherichia coli* ATCC 25922, *Enterococcus faecalis* ATCC 29212, *Acinetobacter baumannii* ATCC 19606, *Bacillus subtilis* SCSIO BS01, *Micrococcus Luteus* SCSIO ML01, and methicillin resistant *S. aureus* ATCC 43300) by the broth microdilution method[8,9]. Bacterial strains were grown to achieve an optical absorbance of 0.04−0.06 at 600 nm on a rotary shaker at 37 °C. Then the bacterial cells were diluted 10-fold in Mueller-Hinton broth and were distributed into 96-well microtiter plates, which were supplemented with compounds (**12**, **14−18**, **24−26**, **33**, **35**, **36**) ranging from 128 to 0.25 μg mL$^{-1}$. The minimum inhibitory concentration (MIC) values were measured after 18 h cultivation. Each compound was tested in triplicates with TMP as a positive control.

**Data availability**. Deposition number of crystallographic data for **15** is CCDC 1584739. The authors declare that all data supporting the findings of this study are available within the article and its Supplementary Information and all other data are available from the corresponding authors upon reasonable request.

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

## Acknowledgements

This work is supported in part by the Chinese Academy of Sciences (XDA11030403, QYZDJ-SSW-DQC004), the National Natural Science Foundation of China (41676165, 31630004), Guangdong Province (2015A030308013, GD2012-D01-002). H.-w. L is funded by the National Institutes of Health (GM040541) and Welch Foundation (F-1511). B.C. D. acknowledges the support of CAS-TWAS President's Fellowship. We are grateful to the analytical facilities in SCSIO. We thank Professor Yi Tang and Dr. Mancheng Tang in UCLA for helpful discussions.

## Author contributions

C.H., W.Z., C.F., Q.Z., and C.S.-Y. performed compound isolation, synthesis, and structure determination. C.Y., B.C.D., Y.Z., and X.J. conducted the in vivo genetic and the in vitro biochemical studies. L.Z. performed the X-ray analysis and protein structure modeling. C.H., C.Y., H.-w.L., and C.Z. analyzed the data and wrote the manuscript. C.Z. and H.-w.L. directed the research. C.H. and C.Y. contributed equally to this work.

## Additional information

**Competing interests:** The authors declare no competing interests.

