## [Peer Review File · Nature Communications]

Reviewers' comments:

Reviewer #1 (Remarks to the Author):

In this manuscript, the authors found that the dimerization process in the biosynthesis of fluostatin (FST)-type dimeric natural products is not an enzyme-catalyzed reaction but occurs spontaneously. The authors initially heterologously expressed the *fls* cluster and obtained several new FST derivatives including dimeric and trimeric molecules. They then performed the *in vitro* characterization of the α/β hydrolase FlsH and revealed that the enzyme is responsible for deacylation reactions. In the course of the FlsH study, the authors interestingly found that some of the substrates used for the enzymatic assay undergoes spontaneous dimerization in the absence of FlsH. The authors also revealed the key features of the substrate that could undergo the dimerization. I believe this manuscript is an interesting and important work since it answered an enigmatic question in natural products chemistry how the dimerization occurs and provides an insight into the biosynthesis of many other dimeric natural products. Overall, I support the publication of this paper in Nature Communications after the authors address the following points.

1. The authors tried to express *fls* gene cluster in several *Streptomyces* sp., did the other hosts produce the same type or different type of dimer products?
2. The formation of p-QM intermediate, the nucleophilicity of donors, and spontaneous deacylation should be affected by pH. Therefore, the pH effect of these non-enzymatic reactions should be shown.
3. If dimerization is important for the cytotoxicity of compounds as written in first paragraph of introduction, I think the authors should check the biological activities for the compounds found in this study.
4. Although the catalytic residues (Asp, His, and Asp) for epoxide hydrolase are not conserved in Lom6, this enzyme can catalyze both epoxide hydroxylation and deacylation reactions. Can the authors discuss more about this enzyme?
5. Figure 3 and Supplementary Figure 14 – In Figure 3, both compounds 8 and 9 are well consumed to produce 4 as a product in the presence of FlsH, while it seems almost no reaction occurs in Supplementary Figure 14 despite the higher enzyme concentration. This should be an important figure by which the authors want to compare the reactivity of the three enzymes. The authors thus need to explain the significant difference between the two figures.
6. Page 5, line 103 – What does “a similar core skeleton” mean? It could be read as 18 also has the SEK 43 moiety.
7. Page 7, line 145 – The sequence identity between FlsH and the proteins used for the model construction should be given.

Reviewer #2 (Remarks to the Author):

The manuscript by Huang and coworkers described non-enzymatic dimerization mechanism to yield benzofluorene-containing angucycline dimers. Benzofluorene-containing angucycline dimers, such as difluostatin A and lomaiviticin A, have unique polycyclic structure possibly biosynthesized from a dimerization of the corresponding monomer unit. To elucidate the dimerization mechanism, the authors examined a series of experiments including 1) heterologous expression of the biosynthetic genes (*fls*) in three different *Streptomyces* hosts and 2) *in vitro* enzymatic reaction

with FlsH. During the course of these experiments, the authors found unexpected deacylation-initiated non-enzymatic dimerization pathway leading to a series of angucycline dimers from acyl-fluostatin derivatives, such as compounds 7 and 8. Involvement of para-quinone methide-like intermediate is supported by the reactions utilizing corresponding methylether derivatives 27 and 28. These results figure out the missing route leading to structurally unique angucycline dimers.

The experiments are scientifically sound and the work is potentially interesting to the researchers of natural product community. However, there are some questions and missing data that the authors may need to address and add to the revised manuscript regarding the proposed non-enzymatic dimerization. Answering the following two major questions would strengthen the proposal.

Major questions

1. Para-quinone methide formation (P8, line 178)

The reviewer has following questions regarding the para-quinone methide formation; 1) Is the polarity of the solvent important for the para-quinone formation? What happens when utilizing non-polar solvents? 2) What is a reason why dimerization occurs only in the H₂O solvent (Figure 5a (iii) vs (vii) for 7/(vi) vs (viii) for 8)? 3) Is it possible to detect the para-quinone methide as in the case of 3,5-di-tert-butyl-4-hydroxybenzyl benzyl acetate (ref 35)? These validations will provide insight into the dimerization mechanism.

2. Site-selectivity of the nucleophile (P10, line 209)

This reviewer thinks that the site-selectivity of the nucleophile might correlate with the HOMO and LUMO energy levels. Taken the fact that the non-enzymatic dimerization is an important topic in this work, the authors should discuss the site-selectivity on the basis of the energy of frontier orbitals (HOMO and LUMO).

Minor comments;

1. Page 5, line 91 (also P S5, supplementary information);

The structure of novel compound 13 is unique in that it has a hydrazine moiety with acetyl group. Otherwise, the readers may not evaluate the adequacy of the structure determination due to the lack of critical spectroscopic data to support existence of the hydrazine moiety. It should be added detailed explanations on the structure determination with appropriate experimental data. The readers will be convinced easily if the authors give appropriate references about examples of the compound with the similar structure.

2. Page 7, line 144;

The quality of the homology modeling depends on the sequence similarity with the template structure. However, there are no comments on the similarity between FlsH and the template alpha,beta-hydrolases. To strengthen the proposed catalytic mechanism, the authors should add some comments on them.

3. Page 9, line 193;

What is the meaning of the term "deacyloxylation"?

4. Page 14, line 294;

Detailed conditions (solvents, flow rate, and monitoring) for reversed-phase semi-preparative high performance liquid chromatography should be added in "Methods" section.

5. Page 18, line 388, reference 3;

The page number should be added instead of DOI number.

6. Page 29, figure 6;

The final step for trifluostatin A formation might be an epoxide opening reaction accompanying an olefin formation. However, the reaction described in Figure 6 is a diol formation. Please check it

carefully and revise the scheme.

Reviewer #3 (Remarks to the Author):

This well-written ms concerns the complex biosynthetic pathway to an emerging group of cytotoxic polyketide products, the benzofluorene-containing atypical angucyclines. The best-known of these are the dimeric lomaiviticins, which are being evaluated as highly potent therapeutics. The authors write concisely and their treatment of the background literature is fair, but it would be worthwhile, for the benefit of the more general reader, to insert a sentence about the mechanism of action of lomaiviticins in inducing double-stranded breaks in DNA.

The ms examines a central issue in the formation of the dimeric end-products, namely the nature of the dimerization step, using as their model system the fluostatin (fls) biosynthetic gene cluster on which they have previously published. They use an attractive combination of synthetic chemistry, careful natural product isolation and characterization, and *in vitro* enzymology. The structural characterization of the novel compounds described is well done. There are refreshingly few typos (though, for example, p9 bottom line Hammett would be correct).

The most important claim made by the ms is that previous speculations made by several groups (including their own) as to the identity of a putative dimerase enzyme, based on sequence similarity to a known dimerase in actinorhodin biosynthesis, are wrong. Instead, they propose that dimerization is a chemical process mediated by a reactive para-quinone methide species which serves as a promiscuous acceptor for 1,4 addition. This is a highly interesting finding which rationalises many hitherto puzzling aspects of dimer formation, including the formation of an array of different heterodimeric products *in vivo* when the fls cluster is expressed in different host strains, via both C-C and C-N linkages.

The authors adduce several lines of evidence to support their claim. First, intermediate monomeric acyl-fluostatins (FSTs) *in vitro* undergo slow spontaneous deacyloxyation leading to dimers in the absence of any enzyme; secondly, this reaction is forestalled by the action of purified recombinant FlsH, which acts as a classic hydrolase. They make the plausible suggestion that FlsH may provide a protective mechanism *in vivo* to avoid the build-up of toxic metabolite in the producing strain. They show convincingly the essential role of the hydroxy group at C-6 in FST J in the deacyloxyation reaction.

An obvious experiment to try might have been an in-frame knockout of the gene encoding the candidate dimerase FlsQ1 to see whether monomeric fluostatins accumulate at the expense of dimeric species (and specifically what does accumulate). To my knowledge, this experiment has not yet been tried on lomaiviticins either. Instead, the present authors attempted to express recombinant FlsQ1 and carry out an *in vitro* assay. This was frustrated by the insolubility of the protein. They also attempted *in vivo* bioconversion using FlsQ1 expressed in *E. coli* to convert FST C and into difluostatin A, without success. These are negative results which might bear less interesting alternative interpretations, but the data in this ms, taken as a whole, are convincing and fully consistent with their proposed chemical mechanism for dimerization. The authors further propose (and provide an example) that the versatility of the chemical coupling might be harnessed to prepare novel dimeric compounds.

The results presented are in my opinion likely to be of interest to a wider readership, as well as stimulating other researchers to continue to unravel the intriguing and highly novel biosynthetic enzymology of these pathways.

Peter Leadlay

Response to reviewers

Reviewer #1 (Remarks to the Author):

In this manuscript, the authors found that the dimerization process in the biosynthesis of fluostatin (FST)-type dimeric natural products is not an enzyme-catalyzed reaction but occurs spontaneously. The authors initially heterologously expressed the fls cluster and obtained several new FST derivatives including dimeric and trimeric molecules. They then performed the in vitro characterization of the α/β hydrolase FlsH and revealed that the enzyme is responsible for deacylation reactions. In the course of the FlsH study, the authors interestingly found that some of the substrates used for the enzymatic assay undergoes spontaneous dimerization in the absence of FlsH. The authors also revealed the key features of the substrate that could undergo the dimerization. I believe this manuscript is an interesting and important work since it answered an enigmatic question in natural products chemistry how the dimerization occurs and provides an insight into the biosynthesis of many other dimeric natural products. Overall, I support the publication of this paper in Nature Communications after the authors address the following points.

Comment 1.1. *The authors tried to express fls gene cluster in several Streptomyces sp., did the other hosts produce the same type or different type of dimer products?*

Response 1.1. We have expressed fls gene cluster in four different hosts, including *S. albus* J1074, *S. coelicolor* YF11, *S. lividans* TK64, and *S. pactum* SCSIO 02999. Different production profiles were observed (Supplementary Fig. 1). While no FST analogues were produced in *S. lividans* TK64 and *S. pactum* SCSIO 02999, different types of FSTs (including different types of dimers) were produced in the other two hosts (Supplementary Fig. 1). Difluostatin A (**6**) was observed in *S. coelicolor* YF11 (ref 8) but not in *S. albus* J1074 (Supplementary Fig. 1). In contrast, difluostatins B-D (**15-17**) and trifluostatin A (**18**) were isolated from *S. albus* J1074 (Fig. 2).

Comment 1.2. *The formation of p-QM intermediate, the nucleophilicity of donors, and spontaneous deacylation should be affected by pH. Therefore, the pH effect of these non-enzymatic reactions should be shown.*

Response 1.2. As suggested by the reviewer, we have tested the pH effects on the non-enzymatic deacylation and dimerization reactions. The results are summarized in revised Supplementary Fig. 29, and are also mentioned in the main text of the revised manuscript (p. 10, 1st paragraph),

“(ii) FST D (**7**) is also stable under acidic conditions (almost no change in buffers with pH lower than 4.0), but is readily converted to dimeric FSTs under basic conditions (Supplementary Fig. 29). This is likely due to the ease of deprotonation of both phenol groups at C6 and C7 under basic conditions to facilitate the formation of the p-QM intermediate **I**, which is the precursor for the subsequent dimerization reaction”.

Comment 1.3. *If dimerization is important for the cytotoxicity of compounds as written in first paragraph of introduction, I think the authors should check the biological activities for the compounds found in this study.*

Response 1.3. According to the reviewer's suggestion, compounds isolated and synthesized in this work were tested for antimicrobial activity against seven indicator strains. The results are summarized in revised Supplemented Table 11 (newly added). Unfortunately, no significant antibacterial activity was found for these compounds. A new sentence reporting this finding has been added to the end of the paragraph right before "Discussion" (p. 11) and reads as,

"Unfortunately, none of the isolated new products showed significant antimicrobial activities against seven indicator strains, including *Staphylococcus aureus* ATCC 29213, *Escherichia coli* ATCC 25922, *Enterococcus faecalis* ATCC 29212, *Acinetobacter baumannii* ATCC 19606, *Bacillus subtilis* SCSIO BS01, *Micrococcus Luteus* SCSIO ML01, and methicillin resistant *S. aureus* ATCC 43300, with MIC (minimal inhibition concentration) values greater than 16 $\mu\text{g mL}^{-1}$ (Supplementary Table 11)."

Comment 1.4. *Although the catalytic residues (Asp, His, and Asp) for epoxide hydrolase are not conserved in Lom6, this enzyme can catalyze both epoxide hydroxylation and decaylation reactions. Can the authors discuss more about this enzyme?*

Response 1.4. [Redacted and the below statement has been revised]

Unfortunately, we don't have access to the substrate for Lom6, epoxykinamycin, and thus cannot conduct assays on epoxide hydrolytic activity. Hence, we prefer not to discuss the dual function of Lom6 in this paper due to lack of experimental support.

[Figure Redacted]

Comment 1.5. *Figure 3 and Supplementary Figure 14 – In Figure 3, both compounds 8 and 9 are well consumed to produce 4 as a product in the presence of FlsH, while it seems almost no reaction occurs in Supplementary Figure 14 despite the higher enzyme concentration. This should be an important figure by which the authors want to compare the reactivity of the three enzymes. The authors thus need to explain the significant difference between the two figures.*

Response 1.5. The difference of the apparent enzyme activity between the two Figures (Fig. 3 and Supplementary Fig. 14 in the last submission) is caused by the aging of the enzymes used in these assays. The enzyme used in Fig. 3 was freshly prepared. However, the enzymes used in the original Supplementary Fig. 14 had been stored at -80 °C for a while. In this experiment, we aimed to examine if the three enzymes could utilize multiple substrates with varied acyl chains. We should have used the same batch of enzymes, but we evidently did not pay enough attention on this matter. We feel sorry for this oversight. For a better comparison of the reactivity of these enzymes, we have now repeated the assays using fresh enzymes and the new results are shown in the revised Supplementary Fig. 15.

Comment 1.6. *Page 5, line 103 – What does "a similar core skeleton" mean? It could be read as 18 also has the SEK 43 moiety.*

Response 1.6. A similar core skeleton means the presence of a 6-membered pyran-ring in compounds **16-18**. To avoid possible confusions, the sentence has been revised to read as,

"A similar 6-membered pyran-ring core also exists in the structure of trimeric trifluostatin A"
(p. 5, last paragraph)

Comment 1.7. *Page 7, line 145 – The sequence identity between FlsH and the proteins used for the model construction should be given.*

Response 1.7. The detailed sequence alignment among FlsH, Lom6 and the protein (PDB ID: 3R0V) used for the model construction is shown as revised Supplementary Fig. 17. We have also added a sub-section “**Protein homology modeling and in silico docking**” in the “Methods” to describe the methods used,

“**Protein homology modeling and in silico docking.** The structure models of FlsH and Alp1U are built by I-Tasser online server⁶⁵. FST J (**8**) is docked into the deduced FlsH active site using AutoDock Vina⁶⁶.” (p. 17)

Reviewer #2 (Remarks to the Author):

The manuscript by Huang and coworkers described non-enzymatic dimerization mechanism to yield benzofluorene-containing angucycline dimers. Benzofluorene-containing angucycline dimers, such as difluostatin A and lomaiviticin A, have unique polycyclic structure possibly biosynthesized from a dimerization of the corresponding monomer unit. To elucidate the dimerization mechanism, the authors examined a series of experiments including 1) heterologous expression of the biosynthetic genes (fls) in three different Streptomyces hosts and 2) in vitro enzymatic reaction with FlsH. During the course of these experiments, the authors found unexpected deacylation-initiated non-enzymatic dimerization pathway leading to a series of angucycline dimers from acyl-fluostatin derivatives, such as compounds 7 and 8. Involvement of para-quinone methide-like intermediate is supported by the reactions utilizing corresponding methylether derivatives 27 and 28. These results figure out the missing route leading to structurally unique angucycline dimers.

The experiments are scientifically sound and the work is potentially interesting to the researchers of natural product community. However, there are some questions and missing data that the authors may need to address and add to the revised manuscript regarding the proposed non-enzymatic dimerization. Answering the following two major questions would strengthen the proposal.

Major questions

Comment 2.1. *Para-quinone methide formation (P8, line 178)*

The reviewer has following questions regarding the para-quinone methide formation; 1) Is the polarity of the solvent important for the para-quinone formation? What happens when utilizing non-polar solvents? 2) What is a reason why dimerization occurs only in the H₂O solvent (Figure 5a (iii) vs (vii) for 7/(vi) vs (viii) for 8)? 3) Is it possible to detect the para-quinone methide as in the case of 3,5-di-tert-butyl-4-hydroxybenzyl benzyl acetate (ref 35)? These validations will provide insight into the dimerization mechanism.

Response 2.1.

1) Acyl FSTs are quite stable in aprotic organic solvents, e.g. dimethyl sulfoxide (DMSO), acetone, and chloroform. Neither deacylation nor dimerization reactions were detected when FST D (**7**) was incubated in these solvents (Supplementary Fig.19). This observation is described in a newly added sentence in the main text to read as,

“The proposed mechanism (Fig. 5b) is further supported by the following facts: (i) acyl FSTs (such as **7**) are quite stable in aprotic organic solvents, e.g. dimethyl sulfoxide (DMSO), acetone, and chloroform (Supplementary Fig. 19)” (p. 10, 1st paragraph)

2) Dimerization also occurs in MeOH (Figure 5a (vii)) for **7**, and the product was identified as difluostatin H (**26**). We suspect that protic solvents could protonate the carboxylate leaving group and thus facilitate the C-O bond cleavage to form the quinone methide intermediate.

3) Unlike in the case of 3,5-di-tert-butyl-4-hydroxybenzyl benzyl acetate (revised ref 36), we were unable to directly detect the formation of para quinone methide from acyl FSTs.

Comment 2.2. *Site-selectivity of the nucleophile (P10, line 209)*

This reviewer thinks that the site-selectivity of the nucleophile might correlate with the HOMO and LUMO energy levels. Taken the fact that the non-enzymatic dimerization is an important topic in this work, the authors should discuss the site-selectivity on the basis of the energy of frontier orbitals (HOMO and LUMO).

Response 2.2. As suggested by the reviewer, a short discussion about the site-selectivity of nucleophilic addition to *p*-QM has been added to p. 12, mid of 2nd paragraph.

“The nucleophilic coupling to a *p*-QM is governed by the HOMO-LUMO interactions between the incoming nucleophile and the quinone methide. The commonly observed δ -addition of nucleophile to the *p*-QM system is both kinetically and thermodynamically more favorable than the β -addition⁵⁴.” (p. 12, 2nd paragraph)

Minor comments

Comment 2.3. *Page 5, line 91 (also P S5, supplementary information);*

The structure of novel compound 13 is unique in that it has a hydrazine moiety with acetyl group. Otherwise, the readers may not evaluate the adequacy of the structure determination due to the lack of critical spectroscopic data to support existence of the hydrazine moiety. It should be added detailed explanations on the structure determination with appropriate experimental data. The readers will be convinced easily if the authors give appropriate references about examples of the compound with the similar structure.

Response 2.3. The acetylhydrazine moiety exists not only in compound **13**, but also in difluostatin B (**15**) and difluostatin D (**17**). Their structures were determined based on extensive NMR analysis. The structure of difluostatin B (**15**) was further confirmed by X-ray analysis (see Fig. 2). A careful comparison of the NMR data of **13** and **17** to those of **15** clearly revealed the presence of an acetylhydrazine moiety in the former two compounds. To provide more details, the text on the structure elucidation of **13** and **17** in p. S5 of the Supplementary Information has been revised to read as,

“The chemical shifts of the quaternary carbons at C5 (δ_C 143.1) and C15 (δ_C 160.0) in compound **13** (Supplementary Table 3, Supplementary Fig. 4) and C5 (δ_C 144.3) and C15 (δ_C 160.3) in compound **17** (Supplementary Table 4, Supplementary Fig. 8) match well with those of their counterparts in compound **15** (C5' δ_C 140.5 and C15' δ_C 168.0), Supplementary Table 4, Supplementary Fig. 6). These data strongly support the presence of an acetylhydrazine (CH₃CONHNH-) moiety at C5 in compounds **13** and **17**.”

Comment 2.4. *Page 7, line 144;*

The quality of the homology modeling depends on the sequence similarity with the template structure. However, there are no comments on the similarity between FlsH and the template alpha,beta-hydrolases. To strengthen the proposed catalytic mechanism, the authors should add some comments on them.

Response 2.4. The structure model of FlsH was built on the template alpha,beta-hydrolase from *Sphaerobacter thermophilus* DSM 20745 (PDB ID: 3R0V), which shares 45% identity for 98% covered sequences with FlsH (see revised Supplementary Fig. 17). Further comparison of the FlsH model with template crystal structures listed by I-Tasser (PDB IDs: 5EGN and 4UHC3) reveals a similar overall alpha/beta hydrolase geometry (RMSD in between 1.0-2.7 Å) and the

presence of a conserved catalytic triad. To reflect this fact, we have revised the main text to read as,

“The structure model of FlsH is built according to a template structure of an α/β hydrolase from *Sphaerobacter thermophilus* DSM 20745 (PDB ID: 3R0V), which shares 45% sequence identity to FlsH (Supplementary Fig. 17).” (p. 7, mid of 2nd paragraph)

Comment 2.5. Page 9, line 193;

What is the meaning of the term “deacyloxylation”?

Response 2.5. The term “deacyloxylation” means “the leaving of an acetoxy group, RCOO⁻”. As shown in the following Figure. “deacyloxylation” of acyl FSTs produces a pQM intermediate, while “deacylation” of acyl FSTs would generate **4**.

Comment 2.6. Page 14, line 294;

Detailed conditions (solvents, flow rate, and monitoring) for reversed-phase semi-preparative high performance liquid chromatography should be added in “Methods” section.

Response 2.6. We have added the conditions of HPLCs to “Methods” to read as,

“The reversed-phase semi-preparative HPLC was performed on a Hitachi-L2130 station (equipped with a Hitachi L-2455 diode array detector) using a Phenomenex C18 column (250 mm \times 10 mm, 2.5 mL min⁻¹) under the following program: solvent system (solvent A, 10% acetonitrile in water supplementing with 0.08% formic acid; solvent B, 90% acetonitrile in water); 5% B to 80% B (0–20 min), 80% B to 100% B (20–21 min), 100% B (21–24 min), 100% B to 5% B (24–25 min), 5% B (25–30 min); flow rate at 2.5 ml min⁻¹, UV detection at 304 nm.” (p. 14, last paragraph)

Comment 2.7. Page 18, line 388, reference 3;

The page number should be added instead of DOI number.

Response 2.7. The page number of reference 3, “7674” (actually the article number in *Nat. Commun.*) has been added to replace “DOI: 10.1038/ncomms8674”.

Comment 2.8. Page 29, figure 6;

The final step for trifluostatin A formation might be an epoxide opening reaction accompanying an olefin formation. However, the reaction described in Figure 6 is a diol formation. Please check it carefully and revise the scheme.

Response 2.8. As highlighted in the following scheme, “dehydration” has been removed and “epoxy opening” changed to “deprotonation and epoxide opening” (see revised Fig. 6). The olefin in trifluostatin A may be formed via an intermediate, which undergoes deprotonation-triggered epoxide opening under basic conditions. A good reference (revised ref 63) is: Tan DS, Dudley GB, Danishefsky SJ. Synthesis of the functionalized tricyclic skeleton of

guanacastepene A: a tandem epoxide-opening β -elimination/Knoevenagel cyclization. *Angew. Chem. Int. Ed.* **41**, 2185-2188 (2002).

A sentence was added in the main text (p. 13, 1st paragraph) to highlight this point,

“The olefin in trifluostatin A may be formed via an intermediate, which undergoes a deprotonation-triggered epoxide opening under basic conditions⁶³.” (p. 13, 1st paragraph)

Reviewer #3 (Remarks to the Author):

This well-written ms concerns the complex biosynthetic pathway to an emerging group of cytotoxic polyketide products, the benzofluorene-containing atypical angucyclines. The best-known of these are the dimeric lomaiviticins, which are being evaluated as highly potent therapeutics. The authors write concisely and their treatment of the background literature is fair, but it would be worthwhile, for the benefit of the more general reader, to insert a sentence about the mechanism of action of lomaiviticins in inducing double-stranded breaks in DNA.

Response: We have revised the main text to highlight the mechanism of action of lomaiviticin A. The revised text now reads as,

“A case in point is the C₂-symmetric lomaiviticin A (1) which exhibits cytotoxicity at nano- to picomolar concentrations by inducing double-strand breaks in DNA, and is currently under preclinical evaluation for antitumor treatment.” (p. 3, line 5)

The ms examines a central issue in the formation of the dimeric end-products, namely the nature of the dimerization step, using as their model system the fluostatin (fls) biosynthetic gene cluster on which they have previously published. They use an attractive combination of synthetic chemistry, careful natural product isolation and characterization, and in vitro enzymology. The structural characterization of the novel compounds described is well done. There are refreshingly few typos (though, for example, p9 bottom line Hammett would be correct).

Response: Page 9, line 200, “Hammet” has been changed to “Hammett”. We also carefully checked throughout the manuscript to correct any typo.

The most important claim made by the ms is that previous speculations made by several groups (including their own) as to the identity of a putative dimerase enzyme, based on sequence similarity to a known dimerase in actinorhodin biosynthesis, are wrong. Instead, they propose

that dimerization is a chemical process mediated by a reactive para-quinone methide species which serves as a promiscuous acceptor for 1,4 addition. This is a highly interesting finding which rationalises many hitherto puzzling aspects of dimer formation, including the formation of an array of different heterodimeric products *in vivo* when the *fls* cluster is expressed in different host strains, via both C-C and C-N linkages.

The authors adduce several lines of evidence to support their claim. First, intermediate monomeric acyl-fluostatins (FSTs) *in vitro* undergo slow spontaneous deacyloxyation leading to dimers in the absence of any enzyme; secondly, this reaction is forestalled by the action of purified recombinant FlsH, which acts as a classic hydrolase. They make the plausible suggestion that FlsH may provide a protective mechanism *in vivo* to avoid the build-up of toxic metabolite in the producing strain. They show convincingly the essential role of the hydroxy group at C-6 in FST J in the deacyloxyation reaction.

An obvious experiment to try might have been an in-frame knockout of the gene encoding the candidate dimerase FlsQ1 to see whether monomeric fluostatins accumulate at the expense of dimeric species (and specifically what does accumulate). To my knowledge, this experiment has not yet been tried on lomaiviticins either. Instead, the present authors attempted to express recombinant FlsQ1 and carry out an *in vitro* assay. This was frustrated by the insolubility of the protein. They also attempted *in vivo* bioconversion using FlsQ1 expressed in *E. coli* to convert FST C and into difluostatin A, without success. These are negative results which might bear less interesting alternative interpretations, but the data in this ms, taken as a whole, are convincing and fully consistent with their proposed chemical mechanism for dimerization. The authors further propose (and provide an example) that the versatility of the chemical coupling might be harnessed to prepare novel dimeric compounds.

The results presented are in my opinion likely to be of interest to a wider readership, as well as stimulating other researchers to continue to unravel the intriguing and highly novel biosynthetic enzymology of these pathways.

Response: The *flsQ1* knockout suggestion is a good one. The knockout of *flsQ1* was thus performed by insertional mutagenesis and the production profile of the resulting mutant was compared to that of the wild type strain. The results are provided in revised Supplementary Fig. 11 revealing the production of difluostatin A (**6**) in the $\Delta flsQ1$ mutant in a trace amount. Our data now clearly show that FlsQ1 is not necessary for the dimer formation. A new section reporting the *flsQ1* gene disruption experiment has been added to the “Method” (see p.15).

“**Insertional mutagenesis of *flsQ1*.** The *flsQ1* gene was inactivated by PCR-targeting method using the apramycin resistance cassette according to our previous study. Details for the insertional inactivation of *flsQ1* are described in Supplementary Fig. 11. Conjugation between *E. coli* ET12567/pUZ8002/pCSG5017 (Supplementary Table 1) to *M. rosaria* SCSIO N160 was performed as previously described. Specifically, the cell pellets of *E. coli* ET12567/pUZ8002/pCSG5017 were gently mixed with mycelia of *M. rosaria* SCSIO N160, and the mixtures were then plated on the ISP4 solid medium. After incubation at 30 °C for 20–24 h, the plates were supplemented with the antibiotics apramycin (100 $\mu\text{g mL}^{-1}$) and trimethoprim (TMP, 100 $\mu\text{g mL}^{-1}$) for the selection of positive transconjugants.”

REVIEWERS' COMMENTS:

Reviewer #1 (Remarks to the Author):

The authors have adequately addressed the reviewer's comments and suggestions. Now, the manuscript is appropriate for publication.

Reviewer #2 (Remarks to the Author):

The comments by authors respond to requests and enrich the manuscript. The paper would be accepted for publication in this journal.

Reviewer #3 (Remarks to the Author):

I am satisfied that the authors have appropriately responded to the referees' comments.

Peter Leadlay